# New Approaches to the Management of Cardiovascular Risk Associated with Sleep Respiratory Disorders in Pediatric Patients

**DOI:** 10.3390/biomedicines12020411

**Published:** 2024-02-09

**Authors:** Esther Solano-Pérez, Carlota Coso, Sofía Romero-Peralta, María Castillo-García, Sonia López-Monzoni, Alfonso Ortigado, Olga Mediano

**Affiliations:** 1Sleep Unit, Pneumology Department, Hospital Universitario de Guadalajara, 19002 Guadalajara, Spain; 1399esther@gmail.com (E.S.-P.); carlootacs@gmail.com (C.C.); sofiamp10@hotmail.com (S.R.-P.); mariacastillogarcia37@gmail.com (M.C.-G.); sonialopezmonzoni@gmail.com (S.L.-M.); 2Centro de Investigación Biomédica en Red de Enfermedades Respiratorias (CIBERES), 28029 Madrid, Spain; 3Instituto de Investigación Sanitaria de Castilla la Mancha (IDISCAM), 45071 Toledo, Spain; 4Sleep Research Institute, 28036 Madrid, Spain; 5Medicine Department, Universidad de Alcalá, 28805 Madrid, Spain; aortigado@secardiologia.es; 6Paediatric Department, Hospital Universitario de Guadalajara, 19002 Guadalajara, Spain

**Keywords:** cardiovascular, echocardiography, sleep apnea, adenotonsillectomy, children

## Abstract

Exposure to risk factors in youth can exacerbate the development of future cardiovascular disease (CVD). Obstructive sleep apnea (OSA), characterized by repetitive episodes of airway obstructions, could trigger said CVD acting as a modifiable risk factor. Measurements from echocardiography have shown impairments in the anatomy and function of the heart related to the severity of OSA. Therefore, the aim of this review was to propose a new clinical approach to the management of cardiovascular risk (CVR) in children based on treating OSA. The review includes studies assessing echocardiographic parameters for cardiac function and structure in pediatric OSA diagnosed using the apnea–hypopnea index (AHI) ≥ 1/h using polysomnography (PSG) and conducted within a year. Based on the reviewed evidence, in addition to PSG, echocardiography should be considered in OSA children in order to indicate the need for treatment and to reduce their future CVR. A follow-up echocardiography after treatment could be performed if impairments in the anatomy and function were found. Prioritizing parameters intimately connected to comorbidity could propel more effective patient-centered care. In conclusion, a reevaluation of pediatric OSA strategies should be considered, emphasizing comorbidity-related parameters in the cardiovascular field. Further studies are needed to assess this approach, potentially leading to enhanced protocols for more effective pediatric OSA treatment and CVR prevention.

## 1. Introduction

### 1.1. Cardiovascular Disease in the Pediatric Population

Cardiovascular disease (CVD) is the main cause of death globally [1]. Although these adverse events are infrequent in children, the basis of these CVDs, atherosclerosis, may begin in childhood [2,3,4]. The exposure to risk factors and behaviors in youth can exacerbate its development. Hence, both addressing the social, economic, and environmental determinants of health to prevent the onset of risk factors (primordial prevention) and intervening to prevent risk factors from progressing into clinical diseases in adulthood (primary prevention) should be contemplated [2,5].

Focusing on primary prevention, some of these factors are related to life habits, while others are hereditary or the result of diseases [6]. The most common factors are excess body mass, high blood pressure (HBP), tobacco, low physical activity, and alterations in glucose and lipid metabolism [3,4,5,7]. Controlling said morbidities could decrease the atherosclerotic process and delay future cardiometabolic disease [8,9,10]. However, the importance of sleep as a possible trigger of CVD is often not considered despite the recent evidence highlighting its association [11,12]. It is imperative to consider the diagnosis and treatment of sleep disorders, even in the absence of clinically manifested CVD, as these may serve as potential contributors to them.

### 1.2. Sleep Disordered Breathing in the Pediatric Population

Sleep disturbances, and more concisely, sleep disordered breathing (SDB), are prevalent conditions in the pediatric population. SDB ranges from primary snoring (PS) to obstructive sleep apnea (OSA), being the prevalence of SDB of 10–17% and between 1 and 4% in OSA [13,14,15]. OSA is characterized by repetitive episodes of partial (hypopnea) or total (apnea) airway obstructions, driving immediate consequences that include changes in intrathoracic pressure, intermittent hypoxia, and sleep fragmentation [16]. The gold standard test to diagnose OSA is polysomnography (PSG), an objective sleep study that collects neurophysiological and cardiorespiratory variables, which is performed in-laboratory during the night. OSA in children is considered when the apnea–hypopnea index (AHI), a parameter obtained from the sleep study that collects the number of respiratory events per hour of sleep, is greater than 1–3 events per hour. Adenotonsillectomy (AT) is the first line and effective treatment in moderate-to-severe OSA patients (AHI ≥ 5) when adenotonsillar hypertrophy, the most-frequent cause in children, is also present [16,17].

### 1.3. OSA Treatment as a Modifiable Cardiovascular Risk Factor

OSA has particularly been related to adverse cardiovascular (CV) responses due to its associated immediate consequences [18], which result in the activation of the sympathetic nervous system, increased oxidative stress, and a proinflammatory and hypercoagulable state [19]. All these processes taking place in children with OSA have an impact on their CV sphere: alterations in hemodynamic and cardiac structure and function, increase in blood pressure (BP) levels with special impact during night, activation of the inflammatory cascade, and dysfunction of endothelium [20]. Therefore, treating OSA could reduce these intermediate mechanisms acting as a modifiable risk factor in the development of future CVD.

In the present review, CV risk (CVR) was evaluated through the alteration of both the anatomy and function of the heart, assessed using the ultrasound-based imaging technique (echocardiography). On the other hand, OSA was considered for the narrative review when AHI ≥ 1/h was identified from PSG performed in-laboratory. Performing an echocardiography as a complementary test to the PSG could help to identify children who have OSA and concomitant increased CVR. Therefore, the main aim of this review was to propose a new clinical management of CVR in children based on treating OSA as a potential modulator of this risk.

## 2. Material and Methods

For the narrative review, Table 1 contains reports including information about echocardiography and PSG within the pediatric population. The terms “echocardiography” and “obstructive sleep apnea” were used in PubMed and Web of Science to search for these articles. The reports had to meet the following criteria: (1) echocardiographic parameters evaluating cardiac function and anatomy; (2) OSA diagnosis using an AHI ≥ 1/h obtained from in-laboratory PSG; (3) realization of PSG and echocardiography within 1 year; (4) age below 18 years; (5) articles written in English from 2000 to the present. Exclusion criteria were as follows: (1) participants with comorbidities, neuromuscular disorders, or syndromic disorders; (2) systematic reviews and meta-analyses; (3) obese-only populations. Finally, after a careful evaluation of all the studies, 10 studies with adequate data meeting the criteria of the authors were integrated into the narrative synthesis.

## 3. Results and Discussion

### 3.1. Association between OSA and Echocardiography Parameters

#### 3.1.1. The Impact of OSA on Cardiac Structure and Function

There is inconsistency in the evidence supporting an association between pediatric OSA and cardiac remodeling. As suggested by previous reviews [21,22], said controversy may be due to heterogeneity in their designs, comprising characteristics of participants (anthropometric variables and medical history), recruitment strategies, follow-up periods, OSA diagnosis and stratification, composition of control groups, echocardiographic parameters, and echocardiographic imaging methods.

OSA in children has been associated with cardiac alterations involving both left (LV) and right (RV) ventricles and pulmonary arteries. Said CV disturbances may be worsened as OSA severity increases but reverted with effective treatment. Related to LV structure impairments associated with OSA, Amin et al., 2002 and 2005 [23,24], found a statistically significant increase in the LV mass associated with OSA severity measured using the AHI, desaturation index (DI), and oxygen saturation (SpO_2_). This hypertrophy improved after treatment. Similarly, Villa et al., 2011 [25], indicated that SDB children (OSA and PS), but not no-SDB participants, had a tendency for increased LV mass. Chan et al., 2009 [26], showed increased relative wall thickness (RWT) and interventricular septal thickness index to height (IVSI) in moderate-to-severe OSA children (AHI > 5/h) vs. mild OSA (AHI 1–5/h). These variables improved in OSA children adequately treated but not in persistent OSA. The same results have been recently published by Domany et al., 2021, in which OSA children (obstructive AHI—OAHI > 1/h) had increased LV mass worsened with OSA severity.

#### 3.1.2. OSA Impact in Cardiac Function

Regarding LV function, the study by Amin et al., 2002 and 2005 [23,24], identified a decrease in LV diastolic function as E/A ratio, which measures flow velocities across the mitral valve, was diminished in PS (AHI < 1/h) and OSA (AHI ≥ 1/h) children. Comparing both groups, this dysfunction only improved in OSA but not in PS after treatment. Similar results were shown by Ugur et al., 2008 [27], in which OSA children (AHI > 2/h) presented diastolic dysfunction demonstrated by an increase in the mitral Em/Am ratio, with changes after AT. Similar results were reported by Chan et al., 2009 [26], demonstrating an alteration in diastolic function with E/e’, a marker of LV filling pressure, which was elevated when the severity of OSA increased. Kaditis et al., 2010 [28], showed decreased LV systolic function in moderate-to-severe OSA (OAHI > 5/h) vs. PS (AHI ≤ 2/h) measured through lower LV ejection fraction (LVEF), defined by the volume ejected in systole in relation to the volume in the ventricle at the end of diastole, and LV shortening fraction (LVSF), a measure of LV contractility. In the study by Villa et al., 2011 [25], LV dysfunction was demonstrated through a diminished LVEF in OSA children (AHI ≥ 1/h) vs. PS (AHI < 1/h), decreased E/A ratio, and increased isovolumetric relaxation time (IVRT). Recently, Domany et al., 2021 [29], reported that, in OSA children (OAHI > 1/h), LV diastolic dysfunction worsened with OSA severity with improvements after AT, measured by a decrease in the E/e’ ratio in the follow-up.

Regarding the association of the RV architecture and OSA, Amin et al., 2002 [23], indicated an association between the OSA and RV dimensions in children independent of BP and body mass, with RV abnormalities being more common in OSA patients (AHI ≥ 1/h) compared to PS (AHI < 1/h). In terms of RV function, Ugur et al., 2008 [27], observed RV diastolic dysfunction through lower tricuspid Em/Am values in the OSA group (AHI > 2/h), with a favorable change after AT, demonstrated by the elevation of this value in the surgery group. In the study by Chan et al., 2009 [26], many measurements of RV dysfunction were significantly different when OSA severity increased: greater RV systolic volume index (RVSVI), lower RV ejection fraction (RVEF), and right ventricular myocardial performance index (RVMPI). Also, an improvement in RVMPI values related to a significant reduction in the AHI was shown after treatment (surgery or nasal steroid therapy). These results were not effective in the group with persistent OSA. Goldbart et al., 2010 [30], found a strong association between the measurement of RV function, tricuspid regurgitation (TR), and OSA severity. Prior to AT, TR values were elevated in comparison to normal values for children, and these measurements normalized after surgery.

Finally, few studies have investigated the effect of OSA on pulmonary arterial pressure (PAP). Ugur et al., 2008 [27], communicated an increase in PAP in OSA children, with a decrease after treatment through surgery. Similarly, Tinano et al., 2022 [31], showed a reduction in pulmonary arterial systolic pressure (PASP) after AT.

However, although most of the studies reported alterations in the CV sphere, other studies have shown that these abnormalities in preoperative echocardiography were not directly associated with OSA severity. Also, these studies did not find a direct association between surgery and cardiac impairments. Kaditis et al., 2010 [28], could not find an association between diastolic function and cardiac structure in SDB. Goldbart et al., 2010 [30], did not find a correlation between the AHI and LV end-diastolic diameter, suggesting a stronger impact on RV than on LV in OSA children with an AHI ≥ 5/h. Finally, Teplitzky et al., 2019 [32], showed no significant functional or structural cardiac impairments in a very severe OSA cohort (AHI ≥ 30/h).

In summary, it has been shown that OSA could trigger cardiac alterations involving LV and RV, highlighting impairments in diastolic and systolic functions, increased mass, and higher PAP. These disturbances, in terms of architecture and function, were usually worsened as OSA severity increased, as measured using the AHI. In many studies, the impact of treatment was assessed, leading to an observation of a general improvement when children were adequately treated.

Therefore, it seems that OSA is an important risk factor in the CV sphere and that OSA treatment may be useful for the health of the heart in this population. The early identification and management of both cardiac issues and sleep disturbances might improve the overall health and quality of life of children, potentially reducing the risk of complications and improving long-term health outcomes.

**Table 1 biomedicines-12-00411-t001:** Summary of previous reports about the association of OSA diagnosis and treatment and CV consequences.

Study (Author, Year)	Type of Study	Number of Participants (Control/OSA)	Age (Years)	Sex (% Males)	OSA Classification	Mean Follow-Up	Other Characteristics	Main CV Outcomes
Tinano MM. et al., 2022 [31]	Retrospective case series study	-/15 (11 AT/4 non-AT)	<10 years	53.3%	OAHI ≥ 1/h Moderate OAHI ≥ 5 < 10/h Severe ≥ 10/h	18.7 months	Brodsky’s grades 3–4 >75% adenoid hypertrophy AT indication	PASP decreased in all AT children (16.6%); The OAHI did not decrease in six AT children (55%) and three non-AT children (75%); Mean min SpO_2_ increased in the AT children (from 87.5% to 90.2%); Clinical improvements were reported despite the persistence of altered OAHI in six children.
Domany KA. et al., 2021 [29]	Combined from 2 prospective longitudinal studies	174/199	5–13 years	59.3%	OAHI > 1/h	6–24 months	Hypertrophy of palatine tonsils who were scheduled for AT	At baseline, OSA children presented LV diastolic dysfunction and increased LV mass, which worsened as OSA severity increased. After AT, there was an improvement in diastolic dysfunction; no similar changes occurred in the controls. OAHI improved following AT, the E/e’ ratio decreased, and e’ increased.
Teplitzky T. et al., 2019 [32]	Case series, retrospective study	47	1–17 years	70.2%	Severe OSA: AHI ≥ 30/h	-	Echocardiographic evaluation within six months prior to AT	Severe OSA children who underwent echocardiographic screening prior to AT did not show significant functional or structural cardiac impairments. Thus, preoperative echocardiographic screening did not identify any abnormalities.
Villa MP. et al., 2011 [25]	Cross-sectional, observational study	21/18 PS/31 OSA	2–16 years	75.7%	PS: AHI < 1/h OSA: AHI ≥ 1/h	-	Echocardiographic examinations were performed in the morning following PSG	A tendency for increased LV mass between the control group and SDB (PS or OSA) was found but was not statistically significant. Significantly higher LVEF values were observed in OSA patients than in patients with PS. SDB patients compared with control subjects presented an alteration in the late phase of LV diastolic function: (1) increase in the A wave amplitude and a reduction in the E/A ratio, although this difference was statistically significant only in patients with PS. (2) statistically higher IVRT values.
Goldbart AD. et al., 2010 [30]	Prospective study	45/90	<3 years	70.4%	AHI ≥ 5/h	3 months	Echocardiography was performed on the morning of AT and 3 months later	The severity of the disease measured using AHI was strongly correlated with the measurements of RV function (TR). Also, TR values were abnormal, and these measurements were normalized before AT. However, no such correlations were seen with LVEDD (diastolic capacity of the heart), suggesting a stronger impact on the RV than on the LV in young children with OSA.
Kaditis AG. et al., 2010 [28]	Cross-sectional study	19/13 PS/14 OSA	2–12 years	52.2%	PS: OAHI ≤ 2/h Mild OSA: OAHI > 2 ≤ 5/h Moderate-to-severe OSA: OAHI > 5/h	-	Echocardiography was carried out the morning after completion of the sleep study;	The main finding was the lower LV systolic function in children with moderate-to-severe OSA compared to subjects with PS (LVEF and LVSF). Diastolic function and cardiac structure indices in the current study were not related to severity of SDB.
Chan JY. et al., 2009 [26]	Community-based study	35/66	6–13 years	73.3%	Mild OSA: AHI ≥ 1 ≤ 5/h Moderate-to-severe OSA: AHI > 5/h	6 months	36 OSA subjects had follow-up assessment: 8 had AT, 9 received nasal steroid therapy, and 19 refused any forms of treatment but agreed to have follow-up assessment	This study documented RV and LV dysfunction and remodeling: RVSVI, RVEF, RVMPI, and E/e’ were significantly different between controls, mild OSA, and moderate-to-severe OSA. RWT and the IVSI were significantly higher in the moderate to severe group compared with the mild group. Only RVMPI, IVSI improved in children with effective treatment. After controlling for age, gender, and BMI z score, children with moderate to severe OSA had a 4.2-fold increased risk of abnormal LV geometry compared with the control group. The cardiac abnormalities improved when treatment for OSA was effective but not in the group with persistent OSA.
Ugur MB. et al., 2008 [27]	Prospective study	26 PS/29 OSA		52.7%	PS: AHI ≤ 2/h OSA: AHI > 2/h	6 months	Echocardiography was performed in the preoperative period and then was repeated in the 6th postoperative month in the OSA group and once in the control group. Adenoidectomy, tonsillectomy, or AT were carried out in 11, 10, and 8 patients, respectively	The results of TDI showed high PAP values in children with adenotonsillar hypertrophy and OSA. There was a remarkable decrease in PAP in the postoperative period. RV diastolic dysfunction was observed, with favorable change after AT, which was demonstrated by the increase in tricuspid Em and Em/Am values in the surgery group. Similarly, mitral Em/Am values increased after surgery, suggesting that LV dysfunction improved postoperatively.
Amin RS. et al., 2005 [24]	Prospective study	15/48	5–18 years	65.1%	PS: AHI < 1/h OSA: -Group 2: AHI ≥ 1 ≤ 5/h -Group 3: AHI > 5/h	1 year	PSG was repeated 8 weeks and 1 year after the initial evaluation; 10 adequately treated OSA children and 10 age- and gender-matched children with PS were recruited for a 1-year follow-up study. Children were managed with different treatments: AT, uvulopalatoplasty, and CPAP	A decrease in the LV diastolic function (decrease in the E/A ratio) was found across the 3 groups. It improved after treatment in the OSA group but not in children with PS. Negative correlation between OSA severity and LV diastolic function was independent of obesity, BP, and LV mass. Increased LV mass index was observed with increased severity of OSA, which improved after the disorder was adequately treated.
Amin RS. et al., 2002 [23]	Cross-sectional study	19 PS/28 OSA	2–18 years	63.8%	PS: AHI < 1/h OSA: AHI ≥ 1/h	-	-	Children with OSA had a statistically significant increased LV mass compared with children with PS. This was correlated with AHI, DI, and lowest SpO_2_. The results of logistic regression (controlling for age, sex, and BMI) indicated that patients with OSA were more likely to have RV (6-fold) and LV (11-fold) abnormalities than patients with PS.

Abbreviation: OSA: obstructive sleep apnea; CV: cardiovascular; AT: adenotonsillectomy; OAHI: obstructive apnea–hypopnea index; PASP: pulmonary arterial systolic pressure; SpO_2_: oxygen saturation; LV: left ventricle; AHI: apnea-hypopnea index; PS: primary snoring; PSG: polysomnography; SDB: sleep disordered breathing; LVEF: left ventricle ejection fraction; IVRT: isovolumetric relaxation time; RV: right ventricle; TR: tricuspid regurgitation; LVEDD: left ventricular end-diastolic diameter; LVSF: left ventricular shortening fraction; RVSVI: right ventricle systolic volume index; RVEF: right ventricle ejection fraction; RVMPI: right ventricular myocardial performance index; RWT: relative wall thickness; IVSI: interventricular septal thickness index to height; BMI: body mass index; TDI: tissue Doppler imaging; PAP: pulmonary arterial pressure; BP: blood pressure; CPAP: continuous positive airway pressure; DI: desaturation index.

### 3.2. Advantages of Performing an Echocardiography Complementary to Polysomnography

In this sense, performing an echocardiography in addition to the sleep study has several advantages. First, it is a short-term, non-invasive, and easy-to-perform technique compared to other procedures such as blood sample analysis used to obtain specific biomarkers described for evaluating cardiac function. Moreover, it is the most used technique in the study of heart disease due to its low cost and its wide availability [33].

Secondly, knowing that OSA has a negative impact on cardiac structure and function, it could be used to identify patients in whom treating OSA with AT could reverse these alterations [24,26,27,29,30,31].

Thirdly, it may help to better characterize OSA patients. Actually, OSA management is based on the AHI, which does not reflect the heterogeneity of the disease [34]. Selecting OSA patients undergoing surgery based on only the AHI could explain the different results found on AT, being the disease resolved in only 60–80% of children [35,36]. Unresolved OSA may not reverse the negative consequences promoted by OSA and could lead to neurocognitive deficits, behavioral changes, low academic performance, and low quality of life, among others [37]. Therefore, in addition to the AHI, new variables including echocardiographic parameters could be considered for better OSA management and define the phenotype of children undergoing AT. In this sense, the echocardiographic study could identify the group of patients in whom, regardless of its severity measured using the AHI, OSA is having an impact on their cardiac function and/or structure. Furthermore, these are potentially reversible changes which could reduce the development of short- and long-term adverse health outcomes. In the same way, follow-up echocardiography and PSG could help monitor the effectiveness of treatment for cardiac conditions and sleep disorders, enabling adjustments to be made when needed [38].

Fourthly, children are an ideal population to perform this technique since their risk factors can be modified before CVD is well established or advanced [20]. Additionally, first-line treatment in said population (AT) is effective in terms of compliance while in adults, first-line treatment, continuous positive airway pressure (CPAP), is conditioned by its adherence. Although the association between OSA and CVD is well known in adults, randomized clinical trials have failed to demonstrate the effect of CPAP in reducing major CV events [39,40,41]. The most sustained hypotheses are the low compliance of treatment and the inclusion of patients with CVD (secondary prevention) in said studies. Thus, selecting the pediatric population could clarify the response to the intervention of OSA in CVR without confounding factors or biases.

Finally, echocardiography is used to identify cardiac abnormalities that can lead to perioperative and postoperative complications in patients with severe OSA, as recommended by the American Thoracic Society (ATS) and American Heart Association (AHA) [42]. Therefore, performing an echocardiography for OSA children treated with AT could prevent said surgery-related difficulties due to cardiac impairments.

Otherwise, echocardiography has several limitations that need to be commented on. The main limitation is the need for qualified personnel who are essential for the realization and interpretation of the results obtained. Both PSG and echocardiography procedures could cause discomfort, particularly in children, which could lead to stress during the tests, potentially affecting the accuracy of the results. Another disadvantage could be the waiting time for both tests to be performed, which might lead to delays in surgery. Finally, echocardiography measures cardiac structure and function but no other important CV alterations such as elevated BP values, which have been related to future CVD in children [39,43].

### 3.3. Management Proposal for Children with CVR and OSA

Historically, the AHI has served as a primary marker guiding surgical indications for values ≥ 5 events per hour of sleep. Recent studies, however, have illuminated structural and functional cardiac abnormalities in children with AHI values below this threshold. Notably, these alterations appear to be potentially reversible with appropriate treatment. In this discourse, we advocate for a paradigm shift in the assessment of treatment indications, favoring a focus on preventing future comorbidities rather than relying solely on parameters that may not encapsulate the entire spectrum of the disease.

Furthermore, emerging data introduce novel parameters within the realm of OSA, which exhibit stronger associations with CV consequences more than the traditional AHI. Consequently, we propose that future clinical guidelines for the OSA definition in children and severity levels should be based on associated comorbidities and their potential ramifications. However, research exploring the genetic basis of OSA and its relationship with CV outcomes has uncovered candidate genes and pathways that may play crucial roles in both conditions. These genetic factors could influence not only the severity and susceptibility to OSA but also contribute to the development and progression of CV comorbidities associated with the disorder [44].

A transformative approach to diagnostic methods is suggested, directing efforts towards parameters more intricately linked to comorbidity. This would not only simplify diagnostic processes but also enhance their relevance in clinical decision-making. We posit that the diagnostic algorithm should incorporate assessments for organic damage, particularly CV impact. Compared to adults, the management of OSA in children faces two relevant problems: (1) the scientific knowledge in OSA and CVR association is less explored and slower in the child population; (2) the incorporation of the results into clinical practice is delayed. In order to mitigate these problems, strategic actions are relevant, including the following: developing research to transfer the knowledge to the pediatric population and the periodic updating of clinical guidelines based on scientific evidence.

Although guidelines published by ATS and AHA recommend performing preoperative echocardiography only on children with severe OSA [42], the reviewed studies have reported that children with SDB from PS to OSA may have cardiac abnormalities. Accordingly, children referred to the sleep unit for suspected SDB should undergo echocardiography in addition to the PSG, to adapt the treatment to the complete resolution of OSA and reduce their CVR. Additionally, a reevaluation of children after treatment may decide future readjustment when necessary, although it is not clear what the ideal follow-up period should be.

Beyond its role in diagnosis, echocardiography serves as a valuable tool for indicating the necessity of treatment and mitigating CVR. By employing echocardiographic parameters that intimately link to comorbidities, such as cardiac structure and function, clinicians can enhance their understanding of the diverse CV manifestations associated. This approach not only aids in treatment decision-making but also in more tailored and patient-centered care strategies. Furthermore, the integration of follow-up echocardiography post treatment, when necessary, provides an additional insight into the effectiveness of interventions, allowing for the continuous refinement of management strategies.

Implementing these modifications would fundamentally alter the current treatment management algorithm, allowing for the identification of patient cohorts deriving clear treatment benefits. This risk-based approach, departing from arbitrary measures, offers a more personalized framework for the management of pediatric OSA. The delay produced in the application of these advances involves the development and implementation of tailored educational programs for healthcare professionals. These programs could focus on disseminating the latest research findings to clinicians to make them well-informed about advancements in the field. Furthermore, fostering collaboration between researchers and healthcare practitioners (specialist and primary care practitioners) through interdisciplinary forums and conferences may enhance mutual understanding and facilitate the translation of research findings into clinical practice also in primary care levels.

As always, in the health management of children, promoting a supportive relationship, addressing parental concerns, promoting understanding, and obtaining informed consent are integral components of the diagnostic and treatment processes. By prioritizing these aspects, healthcare providers can enhance the quality of care and outcomes for pediatric patients.

Finally, the strengths and limitations of this review need to be commented on. The strengths of the study include a summarized broad spectrum of literature that provides a comprehensive overview of echocardiographic parameters in pediatric OSA. The inclusion of rigorous selection criteria, such as the strict age limit (under 18 years), standardized diagnostic criteria (PSG) for diagnosis and classification, and the temporal proximity between both methods, enhances the internal validity of our findings. Moreover, this population is of special interest as it offers insights into a critical developmental stage.

Within the limitations of the study, there is some controversy on the reported results, which may be due to the heterogeneity of the populations, composition of the control group that was predominantly PS children, different follow-up periods for reevaluation, and differences in echocardiographic performance. Additionally, the inclusion criteria for papers may have introduced selection bias, as only studies published in English were included, and those with participants presenting comorbidities or specific disorders were excluded. Additionally, there may be an underrepresentation of certain populations or recent advancements. These limitations should be considered when interpreting the results, and future research could address these constraints to understand the interplay between pediatric OSA and CV health.

## 4. Conclusions

In conclusion, we advocate for a comprehensive reevaluation of definitions, as well as diagnostic and management strategies, in pediatric OSA. By prioritizing parameters intimately connected to comorbidity and organic damage, we can propel the field towards a more effective, patient-centered care.

More studies with an adequate design are necessary to evaluate the possible implementation of this proposed management in the pediatric population. These studies may include a useful test for OSA diagnosis, and echocardiographic parameters should be considered for treatment indication according to its objective impact and reversibility. Therefore, it may prompt the development of novel protocols that could enhance the management of children’s OSA treatment.

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
