# Peer review of "New Approaches to the Management of Cardiovascular Risk Associated with Sleep Respiratory Disorders in Pediatric Patients"

_biomedicines, 2024, doi:10.3390/biomedicines12020411_

Round 1
Reviewer 1 Report
Comments and Suggestions for Authors
1. Title is to ambicious -it is new approach, but not strategy.
2. Many frases are very trivial, for example "Treatment of sleep disorders should be considered even when there are no clinically manifested CVD, since they can be contributors to them."
Or not correct as :.. sleep disordered breathing (SDB), are prevalent conditions in the pediatric population, being this prevalence between 0.7 - 17 %???
3. A lot of slang words and frases, for example "OSA impact in cardiac structure and function" or "relationship between pediatric OSA and cardiac remodeling."
It is not clear, how echocardiography can reflect the heterogeneity of the disease, define the phenotype of children undergoing AT and help to divide patients for groups, which will show better or worse outcome after AT .
The classification of prevention tipes must be corrected and comply with accepted classification (categorized as primal, primary, secondary, and tertiary prevention).
For example, primordial prevention Ruth A. Etzel has described it "all population-level actions and measures that inhibit the emergence and establishment of adverse environmental, economic, and social conditions". This could be reducing air pollution or prohibiting endocrine-disrupting chemicals in food-handling equipment and food contact materials. It is different from authors interpretation.
Comments on the Quality of English LanguageA lot of slang words and frases, for example "OSA impact in cardiac structure and function" or "relationship between pediatric OSA and cardiac remodeling."
Author Response
Olga Mediano
Sleep Unit, Pneumology Dept, Hospital Universitario de Guadalajara.
C/ Donantes de Sangre sn. 19002. Guadalajara, Spain.
+34 949 209 200
olgamediano@hotmail.com
Resubmission of manuscript ID: 2815649 entitled “New approaches in the management of cardiovascular risk associated to sleep respiratory disorders in pediatric patients”
To Biomedicines Editorial Office:
Thank you for your consideration of our manuscript and thank you to the reviewer for the comments. Please find enclosed our responses to the different points raised by the reviewers:
Dear Reviewer 1,
Thank you for your thoughtful review and constructive feedback on our manuscript titled “New approaches in the management of cardiovascular risk associated with sleep respiratory disorders in pediatric patients”. We appreciate the time and effort you invested in providing detailed comments. We have carefully considered your suggestions and have made the necessary revisions to enhance the rigor and clarity of our manuscript. Please find enclosed our responses and changes done as suggested:
1. Title is to ambicious -it is new approach, but not strategy.
To address this issue, we have revised the title to more accurately reflect the nature of our work. The revised title is as follows: “New approaches in the management of cardiovascular risk associated to sleep respiratory disorders in pediatric patients”
2. Many frases are very trivial, for example "Treatment of sleep disorders should be considered even when there are no clinically manifested CVD, since they can be contributors to them."
Trivial phrases have been carefully considered, and we have improved the clarity and significance of our language.
Or not correct as :.. sleep disordered breathing (SDB), are prevalent conditions in the pediatric population, being this prevalence between 0.7 - 17 %???
In reference to the prevalence of sleep-disordered breathing (SDB) in the pediatric population, we have reevaluated the data. The revised statement now accurately reflects the prevalence range, providing a more precise representation of the existing literature. The final paragraph is as follows: “Sleep disturbances, and more concisely, sleep disordered breathing (SDB), are prevalent conditions in the pediatric population. SDB ranges from primary snoring (PS) to obstructive sleep apnea (OSA), being the prevalence of SDB of 10-17% and between 1 and 4% in OSA [13–15].”
3. A lot of slang words and phrases, for example "OSA impact in cardiac structure and function" or "relationship between pediatric OSA and cardiac remodeling."
We acknowledge your concern regarding the use of slang words and phrases in our manuscript. In response, we have conducted a comprehensive language revision to replace any informal or colloquial expressions with more formal and academically appropriate language.
It is not clear how echocardiography can reflect the heterogeneity of the disease, define the phenotype of children undergoing AT and help to divide patients for groups, which will show better or worse outcomes after AT .
We have addressed this concern and provide further clarification. We believe these clarifications enhance the manuscript's comprehensibility regarding the role of echocardiography in characterizing the heterogeneity of pediatric OSA and predicting outcomes after adenotonsillectomy.
The classification of prevention types must be corrected and comply with accepted classification (categorized as primal, primary, secondary, and tertiary prevention).
For example, primordial prevention Ruth A. Etzel has described it "all population-level actions and measures that inhibit the emergence and establishment of adverse environmental, economic, and social conditions". This could be reducing air pollution or prohibiting endocrine-disrupting chemicals in food-handling equipment and food contact materials. It is different from authors interpretation.
Thank you for your feedback. We appreciate your insights on the classification of prevention types. While we understand that the categorization of prevention measures into primal, primary, secondary, and tertiary prevention is widely accepted, we chose to use the terms 'primordial prevention' and 'primary prevention' to emphasize specific aspects of preventive healthcare interventions in the context of our work.
Regarding the definition of primordial prevention, we have expanded upon it to align with the understanding provided by Ruth A. Etzel, which encompasses population-level actions aimed at inhibiting the emergence and establishment of adverse environmental, economic, and social conditions.
We believe that by elaborating on these concepts, the clarity and comprehensibility of the sentence are enhanced, better conveying the importance of preventive measures in addressing risk factors and behaviors in youth. However, we are open to further discussion and refinement if necessary.
Comments on the Quality of English Language
A lot of slang words and frases, for example "OSA impact in cardiac structure and function" or "relationship between pediatric OSA and cardiac remodeling."
We have conducted a comprehensive language revision to replace any informal or colloquial expressions with more formal and academically appropriate language.
Reviewer 2 Report
Comments and Suggestions for Authors
Congratulations on the subject of this study. It is very important. However, its presentation must be strongly improved. Please, my comments in the attached file.

Author Response
Olga Mediano
Sleep Unit, Pneumology Dept, Hospital Universitario de Guadalajara.
C/ Donantes de Sangre sn. 19002. Guadalajara, Spain.
+34 949 209 200
olgamediano@hotmail.com
Resubmission of manuscript ID: 2815649 entitled “New approaches in the management of cardiovascular risk associated to sleep respiratory disorders in pediatric patients”
To Biomedicines Editorial Office:
Thank you for your consideration of our manuscript and thank you to the reviewer for the comments. Please find enclosed our responses to the different points raised by the reviewer:
Dear Reviewer 2,
Thank you for your thoughtful review and constructive feedback on our manuscript titled “New approaches in the management of cardiovascular risk associated with sleep respiratory disorders in pediatric patients”. We appreciate the time and effort you invested in providing detailed comments. We have carefully considered your suggestions and have made the necessary revisions to enhance the rigor and clarity of our manuscript. Please find enclosed our responses and changes done as suggested:
The abstract must be improved. It is necessary to add information about the selection of the papers used in the revision, and add a conclusion.
We have added a brief sentence in the abstract that outlines the criteria and methodology used in the selection of papers for this revision. We have also appended a concluding statement to the abstract to summarize the key findings or implications discussed in the manuscript.
I am suggesting some references that can be important to improve the Introduction and Discussion sections:
Thank you for your recommendation to include additional references in the manuscript. Upon reviewing the six proposed references, we find that while they may be interesting studies, they do not directly pertain to the primary focus of the work, which is centered on cardiovascular issues in pediatric patients without other comorbidities.
While we appreciate the relevance of these studies in other contexts, we believe that maintaining the focus on cardiovascular health in the specific target population is crucial for the coherence and clarity of the manuscript. Therefore, we have decided not to include any additional references beyond those directly related to the main topic and population of interest.
We welcome any further feedback or suggestions you may have regarding the content and scope of the manuscript. Thank you for your understanding and assistance in this matter.
I suggest using "body mass" instead of weight, here, and throughout the manuscript.
We have revised the manuscript to replace instances of the term "weight" with "body mass" in accordance with your suggestion.
I suggest adding some information about the methods used to select the papers used in this narrative review before the results and Discussion section.
We have added a dedicated section before the Results and Discussion sections, outlining the methodology employed in the paper selection process.
Move to a Material and Methods section and to define the method and tools used to access the papers used in this narrative review.
We have added a new section titled "Material and Methods" to the manuscript. This section provides a comprehensive overview of the methodology employed in accessing and reviewing the relevant literature for our narrative review.
Improve the presentation. See all the references.
We have corrected the presentation of the table as suggested.
Add two paragraphs before the Conclusions. 1- with the limitation of the study, 2- with the strengths of the study.
Thank you for your insightful recommendation to include specific paragraphs discussing the limitations and strengths of our study before the Conclusions section. We appreciate your guidance, and we have incorporated the suggested content to enrich the manuscript.
Reviewer 3 Report
Comments and Suggestions for Authors
he paper appeared to be well prepared. There were several points to consider again.
1. Abstract, Discussion, and Conclusion; what are ‘new’ strategies? Please stress them more.
2. The term ‘management’ was used in Title. This reviewer wonders if the other terms such as ‘diagnoses’ are rather proper because the management could include treatments and/or lifestyle modifications.
3. In this narrative review, how or to what extent did the authors consider the ‘quality’ of papers cited? Any rules? It might be described more cautiously.
4. In relation to the abovementioned point, since the cited studies did not have prospective cohort and/or intervention designs, how strong or to what extent could the authors state the management strategies? It might be checked more cautiously.
5. Discussion; the authors could discuss more the inheritance (genetic influences) of OSA and CVD risk.
6. Discussion; the authors might discuss more the relation, concern, understanding and consent of parents in the diagnostic process of children with OSA and CVD risk.
7. Row 238-241; there have been the gap in care of patients between clinical guidelines and research as stated by the authors. The authors might discuss more of the way to mitigate the gap. The discussion might also include the cost effectiveness of real clinics.
8. In relation to the gap, how should we consider the difference in clinical practice between family/general practitioners and specialists?
9. Row 183; the sentence could cite the reference.
10. Row 198; the sentence could cite the reference.
11. Row 210; the sentence could cite the reference.
Author Response
Olga Mediano
Sleep Unit, Pneumology Dept, Hospital Universitario de Guadalajara.
C/ Donantes de Sangre sn. 19002. Guadalajara, Spain.
+34 949 209 200
olgamediano@hotmail.com
Resubmission of manuscript ID: 2815649 entitled “New approaches in the management of cardiovascular risk associated to sleep respiratory disorders in pediatric patients”
To Biomedicines Editorial Office:
Thank you for your consideration of our manuscript and thank you to the reviewer for the comments. Please find enclosed our responses to the different points raised by the reviewer:
Dear Reviewer 3,
Thank you for your thoughtful review and constructive feedback on our manuscript titled “New approaches in the management of cardiovascular risk associated with sleep respiratory disorders in pediatric patients”. We appreciate the time and effort you invested in providing detailed comments. We have carefully considered your suggestions and have made the necessary revisions to enhance the rigor and clarity of our manuscript. Please find enclosed our responses and changes done as suggested:
1- Abstract, Discussion, and Conclusion; what are ‘new’ strategies? Please stress them more.
We have revised the Abstract to provide more explicit emphasis on the novel strategies discussed in the manuscript. In the Discussion and Conclusion sections, we have extended the strategies proposed, providing a more in-depth exploration of their potential impact on the field of pediatric OSA.
2- The term ‘management’ was used in Title. This reviewer wonders if the other terms such as ‘diagnoses’ are rather proper because the management could include treatments and/or lifestyle modifications.
We appreciate your discerning observation and would like to provide clarification on our choice of terminology. The inclusion of the term 'management' in the title was deliberate, aiming to encompass a comprehensive perspective that goes beyond OSA diagnosis. This new approach could have direct implication in the treatment decision and follow up, but adding parameters that indicate or not the need of treatment and the response to it.
3- In this narrative review, how or to what extent did the authors consider the ‘quality’ of papers cited? Any rules? It might be described more cautiously.
We appreciate the concern regarding the methodology description in our narrative review. It's important to note that unlike a systematic review, where detailed methodology is crucial for the replicability and transparency of the study. In a narrative review, the emphasis lies on synthesizing and critically analyzing existing literature rather than on the thoroughness of the search and the application of strict inclusion and exclusion criteria as in a systematic review. Therefore, a detailed methodology description is not a standard requirement in this type of review. However, we have added a new section titled "Material and Methods" to the manuscript. This section provides a comprehensive overview of the methodology employed in accessing and reviewing the relevant literature for our narrative review.
We appreciate your feedback and are committed to improving the quality and presentation of our work in accordance with your suggestions.
4- In relation to the abovementioned point, since the cited studies did not have prospective cohort and/or intervention designs, how strong or to what extent could the authors state the management strategies? It might be checked more cautiously.
We would like to clarify that some of the reviewed studies did, in fact, incorporate prospective designs, including follow-up periods. Where applicable, we have highlighted studies that included post-surgery follow-up data to further emphasize in considering various study designs. However, we have considered that some studies did not have prospective cohort and/or intervention designs, and we have highlighted the importance of interpreting the discussed management strategies with caution due to the nature of the available literature.
5- Discussion; the authors could discuss more the inheritance (genetic influences) of OSA and CVD risk.
We have expanded the discussion to include a more in-depth exploration of the genetic influences that may contribute to cardiovascular risk in individuals with OSA.
6- Discussion; the authors might discuss more the relation, concern, understanding and consent of parents in the diagnostic process of children with OSA and CVD risk.
A specific paragraph highlighting the importance of the involucration of parents in the diagnosis and treatment process in OSA and CV risk prevention has been added to the discussion.
7- Row 238-241; there have been the gap in care of patients between clinical guidelines and research as stated by the authors. The authors might discuss more of the way to mitigate the gap. The discussion might also include the cost effectiveness of real clinics.
We appreciate your recognition of the gap in patient care between clinical guidelines and research. In response to your suggestion, we have expanded the discussion to explore potential strategies for mitigating the identified gap.
8- In relation to the gap, how should we consider the difference in clinical practice between family/general practitioners and specialists?
We appreciate your consideration of this critical aspect and have incorporated relevant considerations. In response to your suggestion, we have expanded our discussion to address the nuanced differences in clinical practice between these healthcare professionals.
9- Row 183; the sentence could cite the reference.
10- Row 198; the sentence could cite the reference.
11- Row 210; the sentence could cite the reference.
We have carefully addressed your specific comments regarding citations in Rows 183, 198, and 210. They all now include a citation to support the information presented.
Round 2
Reviewer 2 Report
Comments and Suggestions for Authors
No additional comments in this revised version. Congratulations, the manuscript was improved.
Author Response
Dear Reviewer 1
Thank you for your suggestions that have improved our manuscript
Reviewer 3 Report
Comments and Suggestions for Authors
The manuscript was largely improved. The database/search engine to select papers for review should be described. The search term and period (by month/year) should be also described. These are important information to guarantee reproducibility of the study.
Author Response
Dear reviewer 3,
Thank you for your comments that have improved our manuscript. The database to select papers, search term and period have been included in the Material and Methods Section as suggested.